# A Ruptured Left Gastric Artery Aneurysm That Neoplasticized during the Course of Coronavirus Disease 2019: A Case Report

**DOI:** 10.3390/pathogens11070815

**Published:** 2022-07-20

**Authors:** Satoshi Ano, Yuto Shinkura, Tsuneaki Kenzaka, Naoaki Kusunoki, Satoru Kawasaki, Hogara Nishisaki

**Affiliations:** 1Department of Internal Medicine, Hyogo Prefectural Tamba Medical Center, 2002-7 Iso, Hikami-cho, Tamba 669-3495, Japan; ano-oka@umin.ac.jp (S.A.); tsumotsumo_ron@yahoo.co.jp (Y.S.); kawasakisa10ru@gmail.com (S.K.); honssk-d@sanynet.ne.jp (H.N.); 2Division of Community Medicine and Career Development, Kobe University Graduate School of Medicine, 2-1-5 Arata-cho, Hyogo-ku, Kobe 652-0032, Japan; 3Department of Radiology, Hyogo Prefectural Tamba Medical Center, 2002-7 Iso, Hikami-cho, Tamba 669-3495, Japan; nkusu0512@yahoo.co.jp

**Keywords:** COVID-19, left gastric aneurysm, abdominal visceral aneurysm formation, medium-sized vessel, case report

## Abstract

Coronavirus disease 2019 (COVID-19) is an acute respiratory syndrome caused by SARS-CoV-2 and is known to cause respiratory and systemic symptoms. A SARS-CoV-2 infection is involved in aneurysm formation, enlargement, and rupture in medium-sized vessels, such as the cerebral and coronary arteries and the aorta. In contrast, its involvement in forming aneurysms in medium-sized vessels other than the cerebral and coronary arteries has not been reported. An 84-year-old Japanese man with COVID-19 was admitted to our hospital. The treatment course was favorable, and the COVID-19 treatment was completed by the 10th day. On day 14, pancreatic enzymes increased mildly. An abdominal computed tomography revealed a ruptured left gastric aneurysm after spontaneous hemostasis. Arterial embolization was performed. In this patient, a new left gastric aneurysm was suspected of having formed and ruptured during the course of the COVID-19 treatment. To the best of our knowledge, this is the first report of abdominal visceral aneurysm formation caused by COVID-19 in a medium-sized vessel, and it is necessary to remember that aneurysms can be formed at any site when treating this syndrome.

## 1. Introduction

Coronavirus disease 2019 (COVID-19) is an acute respiratory syndrome caused by the novel coronavirus (SARS-CoV-2) [1,2]. It is known to cause various extrapulmonary complications, such as multisystem inflammatory syndrome [3], endocrine disturbances [4], and arterial and venous thrombosis [5].

Among the extrapulmonary complications, vascular disorders have been reported to induce the formation, enlargement, and rupture of aortic aneurysms, as well as the enlargement and rupture of existing aneurysms in medium-sized vessels, such as the cerebral arteries [6,7,8]. Additionally, they can present themselves with findings and symptoms similar to those of Kawasaki disease, including the formation of coronary artery aneurysms [9,10]. While there are reports of aneurysm formation in medium-sized vessels, such as the cerebral and coronary arteries, to the best of our knowledge, no studies to date have reported aneurysm formation in medium-sized vessels other than the cerebral and coronary arteries, such as the celiac arteries and arteries beyond the superior and inferior mesenteric arteries.

We present a case of a newly developed left gastric aneurysm during the course of COVID-19 treatment.

## 2. Case Presentation

An 84-year-old Japanese man with an underlying chronic obstructive pulmonary disease visited his family doctor about a fever of 37.3 °C and a malaise lasting for 2 days. He was referred to our hospital as he was tested positive for SARS-CoV-2 antigen. He was not taking any medications regularly. He had been smoking 20 cigarettes a day for 30 years but quit smoking after the age of 50 years.

His vital signs at the time of examination were as follows: GCS, E4V5M6; temperature, 38.4 °C; SpO_2,_ 92% (oxygen 10 L/min); blood pressure, 160/92 mmHg; respiratory rate, 20 times/min; and pulse rate, 91 beats/min. On physical examination, coarse crackles were heard in all lung fields bilaterally. The abdomen was flat and soft, and he had no tenderness. Blood tests revealed a leukocyte count of 5450/µL, a neutrophil level of 84.4%, and a C-reactive protein (CRP) level of 12.5 mg/dL. Arterial blood gas analysis under 10 L/min oxygen administration showed a pH of 7.509, PaO_2_ of 57.8 mmHg, PaCO_2_ of 33.8 mmHg, HCO3^-^ of 26.7 mmol/L, and Lac of 1.2 mmol/L. Other findings are summarized in Table 1. Inflammatory cytokines, such as tumor necrosis factor (TNF)-α, interleukin (IL)-1, and interleukin-6, were not measured.

The chest radiographs showed a cardiothoracic ratio of 55%, sharp bilateral costo-phrenic angles, and frosted shadows in all bilateral lung fields (Figure 1).

Thoracic computed tomography (CT) showed diffuse frosted shadows in both lungs and an infiltrative shadow in part of the right lower lobe (Figure 2).

The SARS-CoV-2 PCR test was positive, and COVID-19 was diagnosed. The patient was hospitalized and was provided appropriate treatment. The patient was treated with dexamethasone (6 mg/day). The remdesivir dose was 200 mg/day on the first day, which was reduced to 100 mg/day on the second and subsequent days. Baricitinib was administered at 4 mg/day, and a high-flow nasal cannula (HFNC) delivering 40 L of oxygen at 70% was placed. Based on the CT findings of the chest, a complication of bacterial pneumonia was suspected, and tazobactam/piperacillin (4.5 g/6 h) was also administered. Blood culture was negative in both sets, and sputum culture showed only oral commensal bacteria. The fever resolved after the second day. The oxygen flow rate through the HFNC was gradually reduced, and the patient was switched to a nasal cannula on the fourth day. The administration of PIPC/T was terminated on the sixth day. The oxygen administration was terminated on the seventh day. Thereafter, no recurrence of fever or decreased oxygenation was observed, and the administration of all drugs for COVID-19 was terminated on the 10th day. In Japan, it is a standard practice not to repeat polymerase chain reaction (PCR) examination after hospitalization for negative confirmation. For this reason, the patient was not followed-up with PCR testing after admission.

His general condition steadily improved. However, on the 13th day of his illness, an elevation in inflammatory markers (leukocytes, 10,720/µL; CRP, 8.52 mg/dL) was observed (Figure 3). On the 14th day, a plain CT scan of the abdomen was performed again due to a mild elevation of pancreatic enzymes, including amylase (149 U/L) and lipase (83 U/L). On the initial admission (Figure 4), gallstones were found at the neck of the gallbladder. However, there was no other significant finding in the dorsal gastric or ventral pancreas. An abdominal ultrasound showed gallbladder stones but there were no findings suggestive of cholecystitis or cholangitis.

However, a re-examination using a plain abdominal CT showed a high density area on the dorsal gastric/pancreatic ventral side, which was suspected to be a fresh hematoma (Figure 4). A contrast-enhanced CT scan of the abdomen on day 16 led to the diagnosis of a ruptured left gastric aneurysm after spontaneous hemostasis (Figure 5).

His vital signs were stable. Arterial embolization was performed on day 20 as a precautionary measure. Initially, we considered coil embolization with isolation and administered a contrast medium through the left gastric artery. However, it was challenging to visualize the peripheral vessels beyond the lesion. Therefore, we decided to use NBCA/lipiodol embolization after determining that the isolation method would be challenging (Figure 6). After the procedure, angiography was also performed from the right gastric artery that could be anastomosed to the embolized lesion. It was confirmed that the blood supply to the left gastric aneurysm had been cut off. A contrast-enhanced abdominal CT performed on day 23 showed the disappearance of blood flow in the aneurysm and a tendency for the intra-abdominal hematoma to disappear. The patient’s postoperative course was favorable, and he was discharged from the hospital on the 25th day after admission. Additional blood tests revealed an IgG level of 1020 mg/dL, an IgG4 level of 45 mg/dL, a proteinase-3-antineutrophil cytoplasmic antibody level of <0.5 IU/mL, and a myeloperoxidase-antineutrophil cytoplasmic antibody level of <0.5 IU/mL.

Six months have passed since his discharge, and he has no abdominal symptoms. The aneurysm has also not recurred.

## 3. Discussion

We present a case of a newly developed left gastric aneurysm during the course of COVID-19 treatment. To the best of our knowledge, this is the first report of a COVID-19-linked aneurysm formation in medium-sized vessels other than cerebral and coronary artery aneurysms. COVID-19 is known to cause extrapulmonary manifestations, and vascular lesions in the abdomen should also be noted.

A visceral artery aneurysm (VAA) is a relatively rare condition. It is reported that the prevalence of VAAs is approximately 1% of the total population, found in 0.01–0.2% of autopsy cases, most of which are detected following rupture [11,12]. The splenic artery is the most common site of occurrence (60%), followed by the hepatic artery, the superior iliac artery, and others. Aneurysms in the gastric and gastroepiploic areas are even rarer among VAAs (approximately 4% of cases) [13]. In the present case, no aneurysm was detected on the plain abdominal CT on admission. However, a dilated left gastric artery was observed on the plain abdominal CT on the 14th day. The same lesion was contrast-enhanced on a contrast-enhanced CT, leading to the diagnosis of a newly formed aneurysm. Arteriosclerosis, intimal degeneration, and inflammation are the primary causes of gastric aneurysms [14]. CT findings in this case showed no evidence of pancreatic enlargement or areas of poor contrast, thus making pancreatitis unlikely. Moreover, the blood tests were negative for IgG4-related disease or vasculitis. Aneurysms have been reported as a rare complication of cholecystitis [15]. However, in this case, there was no evidence of preceding cholecystitis or cholangitis. Owing to the rapid course of the disease, aneurysm formation due to arteriosclerosis or intimal degeneration that occurs with aging was ruled out. Although the aneurysm formed within the COVID-19 treatment period, it was unlikely that it was caused by dexamethasone, remdecivir, or baricitinib as there are no previous reports of aneurysm formation associated with these drugs. Accordingly, we believe that the inflammation caused by COVID-19 triggered the aneurysm formation. Incidentally, a 2021 meta-analysis (32 studies; 10,491 inpatients) found that lymphopenia, thrombocytopenia, as well as elevated D-dimer, elevated CRP, elevated procalcitonin, elevated creatinine, increased aspartate aminotransferase, increased alanine aminotransferase, increased creatinine, and increased LDH levels were significantly associated with ventilation and death [16]. Although most of these items were met in the present case, the direct relationship between these biomarkers and complication rates is not clear. Therefore, future works should examine this issue.

Two mechanisms of inflammatory responses induced by COVID-19 have been reported: one is mediated by angiotensin-converting-enzyme (ACE)-2 receptors [17,18], and the other is independent of ACE-2 receptors [19]. SARS-CoV-2 has an ACE-2 receptor-mediated entry mechanism [17,18]. ACE receptors include ACE-1 and ACE-2 receptors. ACE-1 receptors convert angiotensin (Ang) I to Ang II, and ACE-2 receptors convert Ang II to Ang(1–7). Ang I is converted into biologically active Ang II, which increases blood pressure and promotes inflammation, while Ang(1–7) has an antagonistic effect on Ang II and acts in an inhibitory manner against inflammatory responses. The binding of SARS-CoV-2 to the ACE-2 receptors induces a decrease in ACE-2 receptor activity and suppression of its expression. This indirectly enhances the effect of Ang II, thereby inducing inflammation at the site of infection [17,18]. Furthermore, as ACE-2 receptors are widely expressed not only in the airway and alveolar epithelial cells but also in the intestinal epithelial cells, renal epithelial cells, myocardial cells, and vascular endothelial cells, SARS-CoV-2 infection induces systemic local inflammation via systemic ACE-2 receptors [20]. During this inflammation process, various inflammatory cytokines and transcription factors are activated, one of which is nuclear factor-κB (NF-κB). Normally, NF-κB exists in the cytoplasm in an inactivated state. However, when activated by TNF-α or IL-1, it enters the nucleus and acts as a transcription factor. This induces proinflammatory cytokines, such as TNF-α and IL-6, further promoting inflammatory responses. In contrast, a mechanism that does not involve ACE-2 receptors is the apoptosis-induced pathway in virus-infected cells [19]. Among the molecules leaked from cells during apoptosis are factors, such as high mobility group box 1, adenosine triphosphate, and histones. When pattern recognition receptors recognize these pathogen-associated molecular markers on vascular endothelial cells, NF-κB is activated, and inflammatory cytokines, such as TNF-α, are induced [19], which subsequently enhances the expression of matrix metalloproteinases (MMPs). There are various types of MMPs, and it is believed that these MMPs weaken the arterial wall by degrading fibers such as elastin and collagen that constitute the vessel wall [21,22]. It has been reported that COVID-19 induces the activation of the NF-κB pathway, which in turn induces TNF-α and MMPs [23,24]. In this case, we did not opt for surgical treatment; therefore, pathology was not available. In addition, inflammatory cytokines, such as TNF-α, IL-1, and Il-6, are difficult to measure in general biochemical laboratories in Japan. Therefore, as inflammatory cytokines were not measured, the mechanism of aneurysm formation mediated by them can only be inferred.

COVID-19 can cause extrapulmonary symptoms, and it has been reported that it may increase the risk of enlargement and rupture of abdominal aortic aneurysms [25,26]. There are also reports of COVID-19 being involved in thrombus formation in the celiac artery, the superior mesenteric artery, and the inferior mesenteric artery during a SARS-CoV-2 infection [27,28,29]. In patients with a SARS-CoV-2 infection, attention should be paid not only to pulmonary lesions but also to abdominal lesions. If a patient suddenly complains about abdominal pain or if he or she becomes hemodynamically unstable, rupture of an aneurysm or arterial thrombus should be considered in the differential diagnosis.

## 4. Conclusions

We present a case of a newly developed left gastric aneurysm during the course of COVID-19 treatment. To the best of our knowledge, this is the first case report of aneurysm formation associated with COVID-19 in a medium-sized vessel, especially in the abdominal visceral artery. In patients with a SARS-CoV-2 infection, it is necessary to pay attention not only to pulmonary lesions but also to abdominal aneurysms and arterial thrombi.

## Figures and Tables

**Figure 1 pathogens-11-00815-f001:**
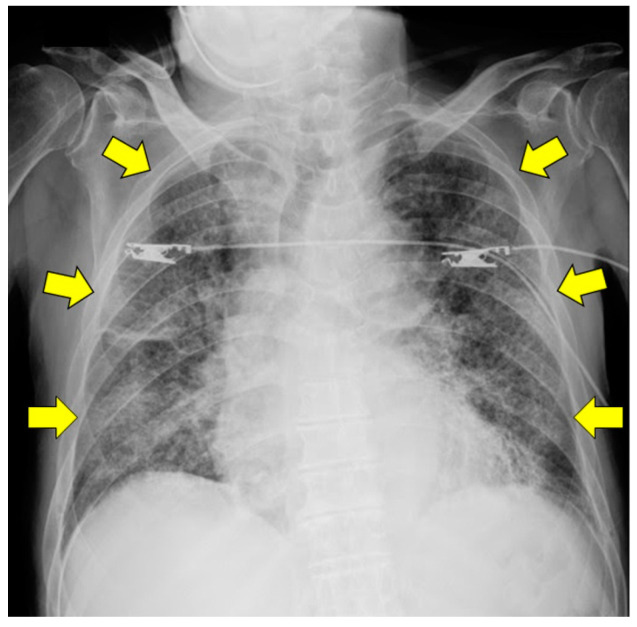
Seated frontal view of the chest radiograph during admission. The whole lung fields on both sides are observed in frosted shadows (yellow arrows).

**Figure 2 pathogens-11-00815-f002:**
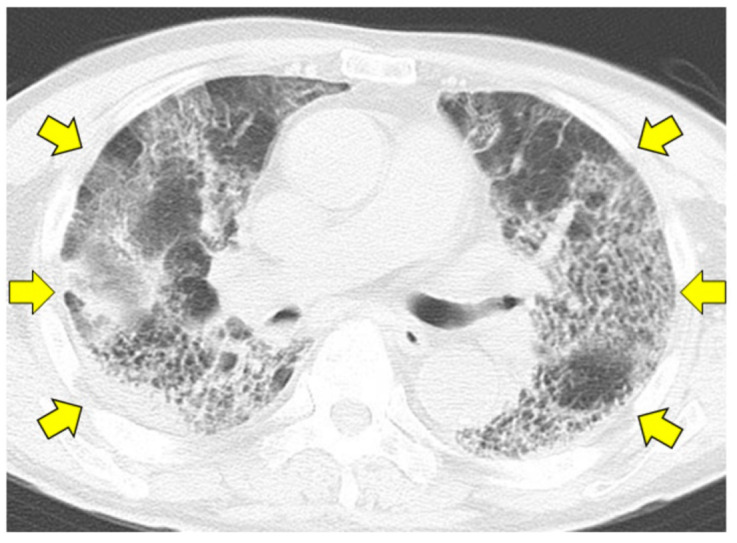
Non-contrast computed tomography of the chest at admission. Diffuse frosted shadows in both lungs and an infiltrative shadow in the right lower lobe can be observed (yellow arrows).

**Figure 3 pathogens-11-00815-f003:**
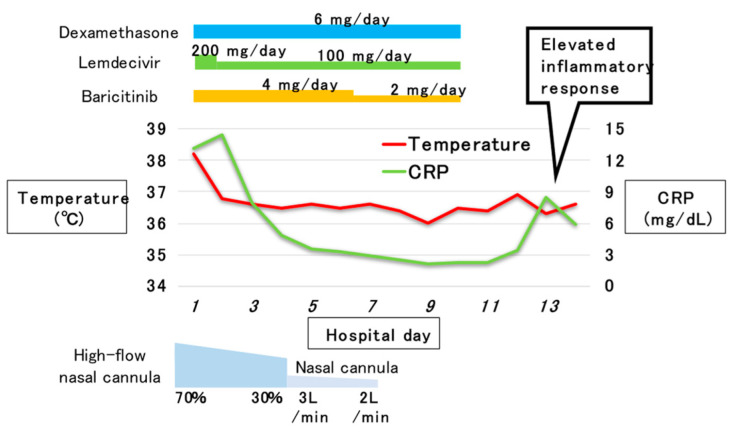
Post-hospitalization course.

**Figure 4 pathogens-11-00815-f004:**
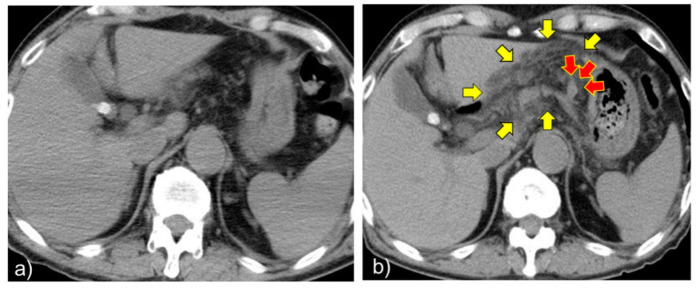
(**a**) Non-contrast computed tomography of the abdomen on admission. Gallstones were found in the neck region of the gallbladder. However, there were no significant findings in the dorsal gastric or ventral pancreas. (**b**) Non-contrast computed tomography of the abdomen on day 14. A highly absorptive zone in the dorsal gastric/peripancreatic region (yellow arrows) and dilatation of the left gastric artery (red arrows) were observed.

**Figure 5 pathogens-11-00815-f005:**
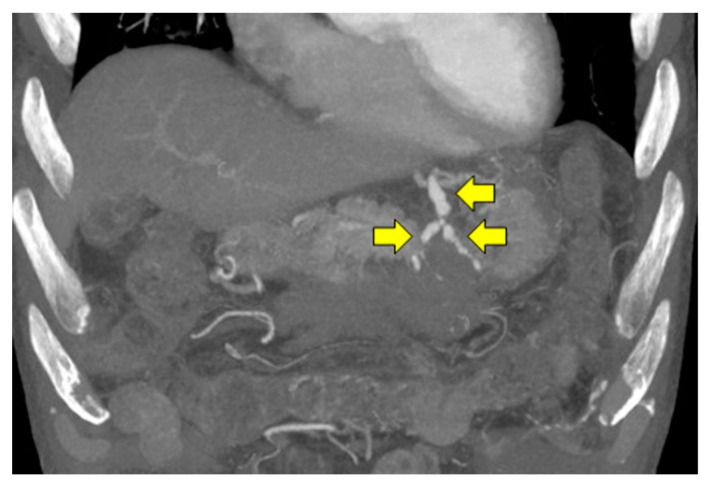
Abdominal contrast computed tomography (Maximum Intensity Projection image) on day 16. A bead-shaped aneurysm formation was seen in the left gastric artery (yellow arrows). The patient was diagnosed after rupture and spontaneous hemostasis of the left gastric artery aneurysm.

**Figure 6 pathogens-11-00815-f006:**
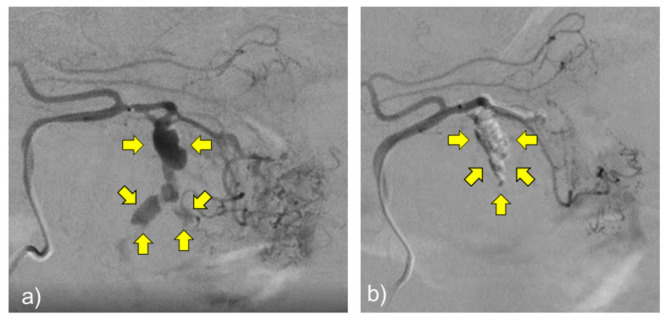
(**a**) Abdominal angiography before arterial embolization. The left gastric artery shows a bead-shaped dilatation (yellow arrows). (**b**) Post-arterial embolization abdominal angiography. Blood flow to the aneurysmal lesion was interrupted (yellow arrows).

**Table 1 pathogens-11-00815-t001:** Laboratory data upon admission.

Parameter	Recorded Value	Standard Value
White blood cell count	5450/µL	4500–7500/µL
Neutrophils	84.4%	42–74%
Lymphocytes	11.4%	18–50%
Monocytes	4.2%	1–10%
Hemoglobin	14.7 g/dL	11.3–15.2 g/dL
Platelet count	9.0 × 10^4^/µL	13–35 × 10^4^/µL
Prothrombin time/ International normalized ratio	0.91	0.80–1.20
Activated partial thromboplastin time	38.7 s	26.9–38.1 s
D-dimer	2.8 μg/mL	≤1.0 μg/mL
C-reactive protein	12.5 mg/L	≤0.60 mg/dL
Procalcitonin	0.14 ng/mL	≤0.05 ng/mL
Total protein	5.4 g/dL	6.9–8.4 g/dL
Albumin	2.5 g/dL	3.9–5.1 g/dL
Total bilirubin	1.1 mg/dL	0.2–1.2 mg/dL
Aspartate aminotransferase	107 U/L	11–30 U/L
Alanine aminotransferase	47 U/L	4–30 U/L
Lactase dehydrogenase	547 U/L	109–216 U/L
Creatine kinase	199 U/L	40–150 U/L
Blood urea nitrogen	16.0 mg/dL	8–20 mg/dL
Creatinine	0.60 mg/dL	0.63–1.03 mg/dL
Sodium	143 mEq/L	136–148 mEq/L
Potassium	2.9 mEq/L	3.6–5.0 mEq/L
Chloride	105 mEq/L	98–108 mEq/L
Glucose	108 mg/dL	70–109 mg/dL
Sialylated carbohydrate antigen KL-6	319 U/mL	≤500 U/mL

## Data Availability

Data sharing is not applicable to this article as no datasets were generated or analyzed during the current study.

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
