# Peer review of "A Ruptured Left Gastric Artery Aneurysm That Neoplasticized during the Course of Coronavirus Disease 2019: A Case Report"

_pathogens, 2022, doi:10.3390/pathogens11070815_

Round 1

Reviewer 1 Report

Dear Authors

The case report is well written and interesting to read. Title is clear and informative; it displays the main objective of the study. The abstract is sectioned. It contains focused background with clear objective. The authors need to expand the review of literature that is relevant to their study. The figures have sufficient, good quality and appropriately illustrative of the paper contents. The case report from a scientific point of view seems to be well done.

The discussion section interprets the findings in view of the results obtained in this and in past studies on this topic.

References cited are recent and have a high relevance to the problem.

Author Response

The authors would like to thank the reviewer for their constructive critique to improve the manuscript. We have made every effort to address the issues raised and to respond to all comments. The revisions are indicated in red font in the revised manuscript. We hope that our revisions meet the reviewer’s expectations.

To support our notion that aneurysms are not caused by anything other than COVID-19, we have presented the results of previous studies and cited the corresponding references. The added part is as follows:

“CT findings in this case showed no evidence of pancreatic enlargement or areas of poor contrast, thus making pancreatitis unlikely. Moreover, the blood tests were negative for IgG4-related disease or vasculitis. Aneurysms have been reported as a rare complication of cholecystitis [15]. However, in this case, there was no evidence of preceding cholecystitis or cholangitis. Owing to the rapid course of the disease, aneurysm formation because of arteriosclerosis or intimal degeneration that occurs with aging was ruled out. Although the aneurysm formed within the COVID-19 treatment period, it was unlikely that it was caused by dexamethasone, remdecivir, or baricitinib as there are no previous reports of aneurysm formation associated with these drugs. Accordingly, we believe that inflammation caused by COVID-19 triggered the aneurysm formation. Incidentally, a 2021 meta-analysis (32 studies; 10,491 inpatients) found that lymphopenia, thrombocytopenia as well as elevated D-dimer, elevated CRP, elevated procalcitonin, elevated creatinine, increased aspartate aminotransferase, increased alanine aminotransferase, increased creatinine, and increased LDH levels were significantly associated with ventilation and death [16]. Although most of these items were met in the present case, the direct relationship between these biomarkers and complication rates is not clear. Therefore, future works should examine this issue.” (Lines 157–173)

The cited references are the following:

  1. Saluja, S.S.; Ray, S.; Gulati, M.S.; Pal, S.; Sahni, P.; Chattopadhyay, T.K. Acute cholecystitis with massive upper gastrointestinal bleed: a case report and review of the literature. BMC Gastroenterol. 2007, 7, 12.
  2. Malik, P.; Patel, U.; Mehta, D.; Patel, N.; Kelkar, R.; Akrmah, M.; Gabrilove, J.L.; Sacks, H. Biomarkers and outcomes of COVID-19 hospitalisations: systematic review and meta-analysis. BMJ Evid. Based Med. 2021, 26, 107–108.

Reviewer 2 Report

The Article may be interesting, if the readers would like to assume a real correlation between COVID-19 and aneurysm formation.

A complete lack of markers of SARS-CoV-2 during invasion/trigger inflammation. Table 1 Clinical Lab analyses is not sufficient.

The Authors and medical doctors should be reported in a new schematic Figure the course of pathology in the initial phase showing also the routinely test for SARS-CoV-2 during the day.

As reported by the Authors in lines 153-158 a readers may think that the outcome was caused by the patient's age and clinical treatment.

In all Figures indicate by arrows what is stated in the captions.

Author Response

The Article may be interesting, if the readers would like to assume a real correlation between COVID-19 and aneurysm formation.

A complete lack of markers of SARS-CoV-2 during invasion/trigger inflammation.

Table 1 Clinical Lab analyses is not sufficient.

Response:

The authors would like to thank the reviewer for their constructive critique to improve the manuscript. We have made every effort to address the issues raised and to respond to all comments. The revisions are indicated in red font in the revised manuscript. Please, find next a detailed, point-by-point response to the reviewer's comments. We hope that our revisions meet the reviewer’s expectations.

Please note that we have added the following part to the revised manuscript:

“Incidentally, a 2021 meta-analysis (32 studies; 10,491 inpatients) found that lymphopenia, thrombocytopenia as well as elevated D-dimer, elevated CRP, elevated procalcitonin, elevated creatinine, increased aspartate aminotransferase, increased alanine aminotransferase, increased creatinine, and increased LDH levels were significantly associated with ventilation and death [16]. Although most of these items were met in the present case, the direct relationship between these biomarkers and complication rates is not clear. Therefore, future works should examine this issue.” (Lines 166–173)

Reference

  1. Malik, P.; Patel, U.; Mehta, D.; Patel, N.; Kelkar, R.; Akrmah, M.; Gabrilove, J.L.; Sacks, H. Biomarkers and outcomes of COVID-19 hospitalisations: systematic review and meta-analysis. BMJ Evid. Based Med. 2021, 26, 107–108.

 “KL-6,” which was within the reference value, was added to Table 1.

Moreover, inflammatory cytokines, such as TNF-α, IL-1, and IL-6, are difficult to measure in general biochemical laboratories in Japan. Therefore, as inflammatory cytokines were not measured, the mechanism of aneurysm formation mediated by them can only be inferred. We have added the following parts to the Case Report and Discussion sections:

“Inflammatory cytokines, such as tumor necrosis factor (TNF)-α, interleukin (IL)-1, and interleukin-6, were not measured.” (Lines 61–62)

“In this case, we did not opt for surgical treatment; therefore, pathology was not available. In addition, inflammatory cytokines, such as TNF-α, IL-1, and Il-6, are difficult to measure in general biochemical laboratories in Japan. Therefore, as inflammatory cytokines were not measured, the mechanism of aneurysm formation mediated by them can only be inferred.” (Lines 203–208)

The Authors and medical doctors should be reported in a new schematic Figure the course of pathology in the initial phase showing also the routinely test for SARS-CoV-2 during the day.

Response:

We would like to thank the reviewer for their insightful comment.

Please note that the course of treatment during hospitalization has been presented in Table 2, and inserted the following text in the revised manuscript:

“In Japan, it is a standard practice not to repeat polymerase chain reaction (PCR) examination after hospitalization for negative confirmation. For this reason, the patient was not followed up with PCR testing after admission.” (Lines 91–93)

In addition, we have revised the following part:

“In this case, we did not opt for surgical treatment; therefore, pathology was not available. In addition, inflammatory cytokines, such as TNF-α, IL-1, and Il-6, are difficult to measure in general biochemical laboratories in Japan. Therefore, as inflammatory cytokines were not measured, the mechanism of aneurysm formation mediated by them can only be inferred.” (Lines 202–207)

As reported by the Authors in lines 153-158 a readers may think that the outcome was caused by the patient's age and clinical treatment.

Response:

As per the reviewer’s insightful comment, we have revised the corresponding part as follows:

“CT findings in this case showed no evidence of pancreatic enlargement or areas of poor contrast, thus making pancreatitis unlikely. Moreover, the blood tests were negative for IgG4-related disease or vasculitis. Aneurysms have been reported as a rare complication of cholecystitis [15]. However, in this case, there was no evidence of preceding cholecystitis or cholangitis. Owing to the rapid course of the disease, aneurysm formation because of arteriosclerosis or intimal degeneration that occurs with aging was ruled out. Although the aneurysm formed within the COVID-19 treatment period, it was unlikely that it was caused by dexamethasone, remdecivir, or baricitinib as there are no previous reports of aneurysm formation associated with these drugs.” (Lines 157–165)

In all Figures indicate by arrows what is stated in the captions.

Response:

We would like to thank the reviewer for the constructive comment. Please note that we have clarified the site by adding arrows.

Round 2

Reviewer 2 Report

The article is improved and the results and rationale is better defined alongside the manuscript.